# Assaying Microglia Functions In Vitro

**DOI:** 10.3390/cells11213414

**Published:** 2022-10-28

**Authors:** Emily Maguire, Natalie Connor-Robson, Bethany Shaw, Rachel O’Donoghue, Nina Stöberl, Hazel Hall-Roberts

**Affiliations:** UK Dementia Research Institute (UK DRI), School of Medicine, Cardiff University, Cardiff CF10 3AT, UK

**Keywords:** microglia, in vitro, functional assays, inflammation, endocytosis, phagocytosis, chemotaxis, motility, immunometabolism, iPSC

## Abstract

Microglia, the main immune modulators of the central nervous system, have key roles in both the developing and adult brain. These functions include shaping healthy neuronal networks, carrying out immune surveillance, mediating inflammatory responses, and disposing of unwanted material. A wide variety of pathological conditions present with microglia dysregulation, highlighting the importance of these cells in both normal brain function and disease. Studies into microglial function in the context of both health and disease thus have the potential to provide tremendous insight across a broad range of research areas. In vitro culture of microglia, using primary cells, cell lines, or induced pluripotent stem cell derived microglia, allows researchers to generate reproducible, robust, and quantifiable data regarding microglia function. A broad range of assays have been successfully developed and optimised for characterizing microglial morphology, mediation of inflammation, endocytosis, phagocytosis, chemotaxis and random motility, and mediation of immunometabolism. This review describes the main functions of microglia, compares existing protocols for measuring these functions in vitro, and highlights common pitfalls and future areas for development. We aim to provide a comprehensive methodological guide for researchers planning to characterise microglial functions within a range of contexts and in vitro models.

## 1. Introduction

Microglia are brain-resident macrophages and act as the main immune modulator cells of the central nervous system (CNS). In the healthy adult brain they are formed with a small cell body and numerous branched processes or ramifications (Sierra et al., 2016). Microglia, which make up ~0.5–16.6% of the total cell population in the human brain [1], display a great deal of heterogeneity in vivo regarding age, sex, and location within the CNS. This includes variation in cell density, morphology, and function [1,2,3]. These cells, previously thought to be largely quiescent within the healthy adult brain, are now known to have complex roles during both development and in maintaining normal brain homeostasis [4]. Microglia are highly versatile, with the ability to rapidly adapt both their morphology and function in response to environmental cues [5].

During development, microglia have key roles in shaping healthy neuronal networks [6]. This includes secreting factors important for neurogenesis [7], promoting oligodendrocyte survival and differentiation [8,9], initiating programmed cell death of neurons and neuronal precursors [10,11,12,13], engulfing synapses [14,15,16], as well as promoting axonal fasciculation and limiting axon outgrowth [17,18]. Within the adult brain, microglia continue to control neurogenesis and modulate neuronal activity, while also carrying out immune surveillance, mediating inflammatory responses, and disposing of unwanted material [6,19].

In vitro microglia cultures provide an incredibly valuable tool to study the functions of these cells, both in the context of health and disease. In vitro models of microglia can be broadly categorised as immortalised microglial cell lines, primary isolated cell cultures from either rodents, macaque or humans, and induced pluripotent stem cell (iPSC) derived microglia-like cells. Each model has advantages and disadvantages, and none successfully recapitulates all characteristics of adult human homeostatic microglia, as is reviewed elsewhere [20,21]. In brief, in vitro microglia appear more “activated”, secreting a greater number of cytokines when compared with their in vivo counterparts. Furthermore, the transcriptomic signature of in vitro microglia differs significantly from in vivo models. These changes likely arise following a lack of signalling between microglia and other CNS cells, which helps microglia retain a more “homeostatic” phenotype [22,23]. Choosing a cell model for functional assays should be informed by the research questions and availability of equipment and expertise.

## 2. Markers for Microglia

The expression profile of microglia is shaped by the local micro-environment and their unique ontogeny. Although they are considered the ‘macrophages of the CNS’, microglia originate from precursors of the primitive yolk sac via PU.1, IRF8 and CSF1R dependent pathways [24,25]. Commonly microglia are characterised using techniques such as RT-qPCR, flow cytometry or immunofluorescence with a range of validated markers which include; IBA1, CD11B, CD45, CD14, CX3CR1, GPR34, MERTK, C1Q, TREM2, SALL1, GLUT5, GAS6, CD68, TMEM119 and P2RY12, which are summarised in Table 1 [26,27,28]. Although these markers are useful in confirming microglia phenotype, they have also shown to be expressed by macrophages with the exception of TMEM119. This is vital in helping to differentiate microglia from macrophages, and subsequently TMEM119 is unique to microglia in the healthy CNS. Despite this, TMEM119 has been shown to be present on some non-CNS cells such as follicular dendritic cells of the spleen and tonsils. Additional popular microglia markers include IBA-1 and P2RY12. Both of these markers are expressed by microglia and macrophages, with expression of P2RY12 on macrophages shown to be much lower than microglia expression [29]. This indicates that neither of these markers can be used to distinguish between microglia and macrophages [30]. Regardless, IBA-1 is of this IBA-1 is useful for visualizing microglia morphology, as staining extends through the slender protrusions of ramified microglia [31]. Furthermore, IBA-1 expression has been shown to be upregulated during phagocytosis and migration processes due to its interaction with actin, which has led it to be considered a good marker for the initial stages of inflammatory microglia activation [32].

Further confirmation of microglia can be shown through the absence of certain markers. For instance, expression of CD206 is characteristic for periventricular macrophages, choroid plexus macrophages and meningeal macrophages, but not microglia, allowing the cell types to be distinguished from one another [25]. Relative quantification of markers can also be useful in confirming cell phenotype. Most notable is CD45 expression, in which perivascular and infiltrating macrophages show high expression while microglia expression is low [33]. However, expression of CD45 on microglia has been shown to increase in models of inflammation and ageing [34]. Indeed, microglia are highly reactive cells constantly surveying for changes in the micro-environment. Upon detection of a stimulus microglia undergo phenotypic changes that can be characterised using various markers. Most commonly, lysosomal marker CD68 is used to detect phagocytic activity [35]. Other markers including IBA1, CD14, CD45, CD11B and CX3CR1 have been shown to be upregulated in various microglia inflammatory models [36,37]. Contrary to this, slight reductions in TMEM119 and P2RY12 mRNA have been noted in pro-inflammatory microglia [30,38].

It is important to be wary of some marker discrepancies across microglia models, some key changes have been summarised in Table 1. Although no single marker has shown to be exclusively expressed by microglia, multiple proteins are useful in the confirmation of microglia phenotype, most commonly used TMEM119 and P2RY12.

## 3. Morphology

Microglial morphology, first described in pioneering research by Pío del Río-Hortega in 1919 [57], is incredibly fluid, with microglia constantly altering their structure in response to external stimuli [58]. Microglia structure and function are tightly coupled, with structural changes likely rendering microglia more efficient at carrying out whichever action is required [58]. Within the healthy, adult CNS, “homeostatic” microglia present with small round cell bodies surrounded by extensive, branching processes. This morphology is known as ramified, and microglia use their long processes or ramifications to continuously survey their immediate environment [4]. Upon detection of an insult (e.g., damaged synapse, misfolded protein) microglia retract their processes and take on a more amoeboid shape, indicative of “activation”. This “activated” state is accompanied by proinflammatory cytokine secretion and increased phagocytic activity [5]. Microglia morphology can be seen to change drastically during both physiological and pathological ageing, as well as in response to a variety of different neurological conditions [58,59].

Histological staining with various markers can accurately delineate the cytoplasm and processes of microglia, with confocal microscopy often used to obtain high quality Z-stack images of these cells. While both fluorescence and 3,3′-diaminobenzidine can be used to obtain images, fluorescence staining often allows for superior visualization of microglial processes, and is therefore preferred [60]. Fluorescence staining for microglia area often incorporates antibodies against IBA1, although other markers that accurately highlight cell structures can also be used [60].

Having obtained a clear image highlighting microglial structure, a variety of image analysis software can be used to perform semi-automated quantification of morphological characteristics. While in vitro monocultured microglia morphology can be quantified using either 2D or 3D images, 3D analysis reduces the likelihood of cell structure misrepresentation. Imaris is one such software [61]. Using Imaris (Oxford Instruments), microglial surfaces can be reconstructed in 3D from Z-stack images prior to quantification of various morphological characteristics (including cell surface area and volume) [62]. Sholl analysis, which provides a measure describing how ramified the microglia are, can also be performed using Imaris filament reconstruction mode [62]. While incredibly useful when characterising microglial morphology, Imaris is unfortunately a paid software and therefore may not be accessible to all researchers. Image J, a free alternative software, can also be used to characterise microglial structure. Several image J plugins exist which can be used to measure and describe microglial morphology according to metrics such as degree of ramification, cell complexity and cell shape [60]. Sholl analysis can also be performed using Image J [63]. However, when comparing Imaris against several ImageJ analysis pipelines to analyse dendrites within Drosophila models, Imaris appeared to generate data most comparable to that obtained with manual tracing [64]. In addition to these widely used software’s, other scripts exist freely online to quantify microglial morphology, although they may require more advanced programming skills [65,66,67,68,69].

To conclude, an understanding of microglial morphology can act as a powerful tool when looking to understand the biology of these cells, and a variety of different semi-automated software exist to aid morphological quantification. However, researchers should be careful not to infer function from morphology, particularly as microglial morphology is highly diverse and transient [70]. Furthermore, it should be noted that isolated microglia never achieve the morphological complexity of microglia observed in situ [20].

## 4. Neuroinflammation

### 4.1. Overview of Neuroinflammation

Microglia are the major mediators of the inflammatory response in the CNS. Neuroinflammation is initiated following binding of either damage-associated molecular patterns (DAMPs), pathogen-associated molecular patterns (PAMPs), or neurodegeneration-associated molecular patterns (NAMPs) by pattern-recognition receptors (PRRs) on microglia [71,72]. PRRs include but are not limited to Toll-like receptors (TLRs) and inflammasome-forming nucleotide binding oligomerization domain (nod)-like receptors (NLRs) [71]. TLR activation results in NFκB translocation to the nucleus and pro-inflammatory gene expression [73]. NLR receptor binding ultimately leads to formation of the cytosolic multiprotein inflammasome complex, and the production and release of inflammatory cytokines IL-1β and IL-18 [74,75]. Within the large family of NLR receptors, NLRP3, which can be activated by both DAMPs and PAMPs, is most highly expressed on microglia [76].

In brief, binding of ligands to pro-inflammatory PRRs on microglia results in the release of numerous mediators of the inflammatory response including cytokines (e.g., TNF-α, IL-6), chemokines (e.g., MCP-1, RANTES), nitric oxide synthase (NOS2) and reactive oxygen species (ROS) [71]. Following a pro-inflammatory response to stimulus, e.g., bacterial infection, immune resolution is usually achieved by microglia releasing anti-inflammatory cytokines (e.g., TGF-β and IL-10) that inhibit proinflammatory cytokine release and promote tissue regeneration [77]. While the initial pro-inflammatory response is crucial to defend the brain following insult or injury; chronic, sustained pro-inflammatory responses can exacerbate damage, and is a characteristic feature of several diseases [78]. ‘Activation states’, or profiles of microglia gene/protein expression, are highly context-dependent, model-dependent, and disease-dependent. Transcriptomics and proteomic profiling have shown us that there are not just two or three possible activation states, but instead innumerable distinct activation states [79,80].

### 4.2. Stimuli Used in Neuroinflammation Assays

Several stimuli are utilised by researchers in order to induce an inflammatory response prior to characterization. The most well-studied microglial activator in vitro is the bacterial cell wall component lipopolysaccharide (LPS), which acts via TLR4, and is suggested to induce a pro-inflammatory phenotype [81]. However, as most CNS insults to which microglia respond will not be driven by bacterial infection, LPS cannot be considered the most relevant stimuli when examining effects of either neurodegenerative disease, healthy aging, or many other conditions. For this reason, more endogenous pro-inflammatory mediators such as IFNγ plus TNF-α, are being increasingly utilised to study microglial activation [81]. Activation of inflammasome formation via NLRP3 is a two-step process. First, cells are usually primed by using a TLR agonist (e.g., LPS) followed by treatment with an activator (e.g., ATP, nigericin) to complete activation [82,83].

### 4.3. Functional Assays for Microglia-Mediated Neuroinflammation

To assess cytokine or chemokines secreted by microglia in vitro, it is common to collect conditioned cell media to perform assays. A summary of the different techniques used to quantify cytokine release from microglia, as well as pros and cons of each technique, is presented in Table 2. The most commonly used methods to quantify cytokine and chemokines are traditional antibody sandwich enzyme-linked immunosorbent assays (ELISAs) and multiplex arrays [84]. Multiplex arrays allow detection of multiple cytokines and chemokines simultaneously within a single sample. They are usually either chemiluminescence, electrochemiluminescence or bead-based [84]. Chemiluminescence and electrochemiluminescence multiplex assays consist of multiple specific capture antibodies at multiple spots (one antibody per spot), which can then be used to detect multiple cytokines in the same sample at the corresponding spots. With bead-based assays, separate capture antibodies are conjugated to bead sets and multiplexed. A graphical summary further explaining how these assays work can be seen in Figure 1.

ROS and nitric oxide, other key mediators of the microglial inflammatory response, can be measured using several easy to use, relatively cheap, commercially available kits [81,85,86]. These kits provide only limited information, detecting specific radical species such as O^2−^. For more in depth analysis, spin trap techniques and Electronic Paramagnetic Resonance (EPR) spectroscopy can be used. However, EPR spectroscopy is an expensive and highly specialised technique, thus limiting its use in many lab settings [87].

To conclude, when performing in vitro assays for neuroinflammatory effects of microglia, a variety of assays exist that focus on different types of inflammatory effects. Key considerations include choosing the most appropriate stimulus for the research question. The most commonly used stimulus, LPS, may be considered less-physiologically relevant when attempting to model non-bacterial driven neuroinflammation. Furthermore, choosing a relevant stimulus concentration and exposure time can be difficult, due to a lack of consensus in the literature. Future studies comparing the inflammatory reactions of microglia to different immune-relevant stimuli would help the field identify the most physiological stimulus, concentration, and exposure conditions. 

## 5. Endocytosis

### 5.1. Overview of Endocytosis

The endocytic pathway is required for effective sorting and recycling of cellular components and is the mechanism by which cells internalise external or plasma membrane bound cargos. However, the role and impact of the endocytic pathway is far more wide reaching. Like all cells microglia require efficient endocytosis and employ it in the internalisation of nutrients, antigen presentation, motility, lipid homeostasis, synapse pruning and membrane receptor regulation [88,89]. One of the main roles microglia perform is phagocytosis, a specialised form of endocytosis, and this cargo internalised via phagocytosis requires sorting via endocytic compartments. 

Endocytosis can be split into clathrin mediated endocytosis (CME), macropinocytosis and phagocytosis (Figure 2). Each pathway internalises different cargoes and uses distinct molecular regulators. CME, first discovered over 50 years ago [90], allows the internalisation of particles between 60–120 nm in diameter and requires clathrin, dynamin and endophilin as key regulating proteins [91]. Macropinocytosis, first observed in 1931 [92], enables cells including microglia to enclose and ingest large volumes of extracellular medium in a nonselective manner, therefore taking in nutrients and exogenous antigens, and is important in antigen presentation [93,94]. This process is actin-dependent and allows cargo of over 200 nm in diameter to enter the cell [95]. Finally, phagocytosis is used by microglia to uptake foreign particles in large vesicles that can be greater than 500 nm in diameter [96,97]. Microglial phagocytosis will be discussed in more detail in Section 6. 

Once cargoes are internalised, they are then shuttled to the early endosome to begin the intracellular trafficking process. Cargo can then be passed onto either the sorting endosome, where cargoes can be recycled back to the plasma membrane, or to the late endosome, which marks cargos for eventual degradation. 

### 5.2. Functional Assays for Endocytosis

Microglia, like other cell types, can be assessed for endosomal network properties such as number of early/late/recycling endosomes, cellular positioning of endosomes and size of different endosomal compartments using immunocytochemistry or electron microscopy [98,99,100,101,102]. Endosomal network changes can be indicative of endosomal traffic jams or improper sorting [103,104]. Commonly used markers of the early endosome include EEA1 and RAB5, whereas the recycling endosome is marked by RAB11 and the late endosome RAB7 [105]. To give a more dynamic assessment of these processes, fluorescently tagged constructs of these markers have been employed to mark and watch endosomal populations in real time [106]. One consideration when studying microglia is the ability to effectively transfect or transduce them in order to allow expression of such constructs given their phagocytic nature [107,108].

Endocytosis is a dynamic process, and in vitro studies can take advantage of this through the use of live imaging to quantify uptake and turnover of specific cargoes over time. Such cargos can be used to distinguish different forms of endocytosis. For example, CME can be investigated using transferrin or epidermal growth factor labelled with pHrodo or permanent fluorescent dyes, using similar methodology to phagocytosis assays, as will be discussed in more detail later [98,109,110,111,112]. An advantage of using pHrodo labelled bioparticles is that they will only fluoresce in acidic compartments which therefore reduces background fluorescence and indicates inclusion in endosomes. Whereas methods for assessing phagocytosis, as described in Section 6, and CME by cargo specificity is relatively straight forward measuring macropinocytosis is more difficult. Dextrans can be used to help differentiate macropinocytosis only if very large dextrans (2,000,000 Da) are used, concentrations are titered carefully, and limits are set on the size of endocytic structure included in analysis [93,95,113]. Smaller dextran particles can be used to investigate CME. For all endocytic uptake assays, careful consideration of controls should be employed to ensure that the endocytic uptake mechanism is as expected. For example, dynasore or pitstop 2 can be used to block CME by blocking the GTPase activity of dynamin and interfering with clathrin terminal domain, respectively [95,114,115]. Cytochalasin D can be used to depolymerise F-actin and can therefore inhibit both micropinocytosis and phagocytosis, and Amiloride can be used to block macropinocytosis through inhibition of Na+/H+ exchange [116,117].

In neurodegenerative diseases such as Alzheimer’s and Parkinson’s Disease, microglia endocytose specific misfolded proteins such as β-amyloid, tau, and α-synuclein [118,119,120,121]. These aggregated proteins may be prepared as small oligomeric species associated with greatest cytotoxicity [122], or they may be fibrillised into large insoluble fibrillar aggregates, which are associated with characteristic histological pathologies [123]. Such proteins of interest can be labelled with fluorescent tags and uptake quantified through imaging. However, caution should be used when labelling these preparations and proper characterisation of aggregates should be undertaken as it is likely that this alters their folding capabilities. Another consideration when doing such experiments is the form of the aggregate-prone protein assembly that the microglia are exposed to, i.e., monomeric, oligomeric or fibrillar, as this will affect both the mode of uptake and the microglia phenotypic response [118,124,125]. Uptake of β-amyloid and tau oligomers is by clathrin-mediated endocytosis and macropinocytosis [126], as these assemblies have been shown experimentally to be 10–50 nm diameter [127,128]. Large fibrillar aggregates of β-amyloid are phagocytosed by microglia, and can be assayed using fluorescent-labelled recombinant fibrils [123,126]. Phagocytosis is discussed in detail in Section 6.

In conclusion, it is possible to investigate the different endocytic pathways through the careful choice of cargo ensuring the correct size for the pathway of interest. This should always be coupled with control compounds, e.g., dynasore to further validate results. Where aggregate-prone proteins such as β-amyloid are used for endocytic uptake experiments, they require full characterisation so that the experimenter is certain of the species being used, as this will impact the pathway of uptake. 

## 6. Phagocytosis

### 6.1. Overview of Phagocytosis

Phagocytosis is an important function of microglia during development and disease. Phagocytosis is defined as the recognition and ingestion of particles larger than 0.5 µm [129]. Microglia are professional phagocytes and can quickly and efficiently clear up apoptotic cells, cell and myelin debris, aggregated proteins, invading micro-organisms, and complement-tagged synapses of live neurons [130]. The process of phagocytosis is initiated by recognition of the target ‘cargo’ by microglia phagocytic receptors. There is a diverse array of phagocytic receptors, and each selectively recognises a specific molecular pattern on a microorganism or apoptotic cell, or an opsonin molecule coating the target (e.g., IgG, C1q) [129]. Phagocytosis receptors become active only when several cluster together within the plane of the membrane, due to engagement with the target, and this triggers a cascade of intracellular signalling that result in re-organization of the actin cytoskeleton to protrude the plasma membrane around the cargo in a ‘phagocytic cup’ structure or ‘pseudopod’ [129]. The phagocytic cup grows until the leading edges fuse together to fully engulf and internalise the cargo into a ’phagosome’ [131]. The phagosome matures by a sequence of fusion and fission events with early and late endosome, causing increasing acidification of the compartment and the recycling of receptors back to the plasma membrane [131]. Lysosomes fuse with the phagosome to enable the cargo to be degraded by lysosomal hydrolases, and then degradation products are released directly into the cytosol, or secreted into the extracellular space within exosomes [130]. The biology of phagocytosis is reviewed elsewhere in more detail [129,130,131]. Phagocytosis can be assayed in vitro by adding a fluorescent-labelled substrate or “cargo” to the microglia cell culture model, and allowing the cells to engulf the material at 37 °C for a period of time. An end-point assay may be performed at a single time-point, or the phagocytosis events may be detected in the live cells at multiple time-points to allow the real-time kinetics to be studied. Table 3 compares the different types of detection instrument that can used, including references for example assays. An overview of phagocytosis, as well as different phagocytic cargos that can be used in assays, is shown in Figure 2.

### 6.2. Cargo Used in Microglia Phagocytosis Assays

Phagocytosis of different cargo can result in different phenotypic outcomes for the microglia. For example, phagocytosis of bacteria generally leads to pro-inflammatory activation and slow degradation that allows antigens to be preserved for presentation, conversely recognition of apoptotic cells suppresses inflammation and cargo is rapidly degraded [131]. Phenotypic outcomes of phagocytosis may be mainly mediated by phagocytic receptor-mediated signalling [129]. Therefore, cargo-specificity is an important consideration for designing a strategy to experimentally interrogate phagocytosis. Researchers that wish to largely avoid cargo-specific responses may turn to polystyrene or latex beads [132], although the receptors involved with unopsonised bead uptake have not been characterised. Often the use of several cargoes is informative. For example, one study determined that TREM2 missense mutations impaired microglia phagocytosis of apoptotic cells but not *E. coli* or Zymosan (a yeast cell wall glucan) [133].

*E. coli* and Zymosan bioparticles are popularly used for phagocytosis studies due to their being commercially available conjugated to a pH-sensitive dye, offering convenience and homogeneity. *E. coli* is recognised by scavenger receptors such as BAI1 and MARCO, and PRRs such as CD14 and SR-A, whereas Zymosan is primarily recognised by Dectin-1 (CLEC7A) receptors [129]. Zymosan bioparticles are large (approx. 4 μm), enabling easy study of the co-localization of phagosome-associated proteins [134]. The main disadvantage of these cargo is that they are pathogens, and therefore of limited physiological relevance to the study of development and neurodegeneration.

Microglial phagocytosis of apoptotic cells, also known as “efferocytosis”, is important in both development and disease. Apoptotic cells universally expose a phosphatidylserine “eat-me” signal, recognised directly by TREM2 and GPR56 receptors, and indirectly (with specific opsonins) by other microglial receptors including MERTK, MEGF10, αVβ3/5 integrin, LRP1, and complement receptors [135]. A simple model of efferocytosis involves measuring the uptake of carboxylate microbeads, which are believed to bind the same receptors as phosphatidylserine due to the negative charge [136]. Various models of apoptotic neurons have been developed for phagocytosis assays using neuronal cancer cell lines: human SH-SY5Y, murine Neuro-2a and rat PC12 cells. These were killed by protocols including UV-irradiation [137], paraformaldehyde [138], staurosporine [139], okadaic acid [140], or oxygen-glucose deprivation [141].

Myelin debris is also cleared by microglia via phagocytosis. Microglial receptors for myelin include MERTK, AXL and TREM2 [142], and complement-opsonised myelin is recognised by CR3 and SR-A [143]. Myelin debris can be purified by sucrose gradient fractionation of mouse or human homogenised brain tissue, labelled with a fluorescent dye, and applied to microglia cell cultures [144,145].

Microglia phagocytose the presynaptic terminals of viable neurons in a process known as “synaptic pruning”, which is normally highly active during development to refine neuronal circuits, and appears to be reactivated in multiple models of neurological disease [146,147]. Synapse phagocytosis is often assayed in vitro by preparing fluorescent labelled synaptosomes and synaptoneurosomes, and presenting to microglia as cargo. Synaptosomes are resealed presynaptic terminals, which are pinched off during homogenization of neurons and retain some functionality. A small proportion remain attached to a resealed postsynaptic terminal, these are referred to as synaptoneurosomes [148]. Synaptosomes and synaptoneurosomes can be obtained from homogenised rodent brain, human post-mortem brain tissue, or iPSC-derived cortical neurons, and enriched by Percoll or sucrose density gradient fractionation [123,138,149,150]. However, a disadvantage of this method is that no prep is pure, there will be significant contamination with lysed cell membranes, some myelin debris and mitochondria [148]. Therefore, it is questionable whether microglia respond to synaptosome preparations similarly to intact neurons.

### 6.3. General Considerations for Phagocytosis Assay Development

Other important considerations for phagocytosis assay development include the duration of phagocytosis (if not using a kinetic assay) and the ratio of cargo particles to microglia. Phagocytosis needs to be captured at a point in time where the rate of phagocytic uptake is constant, and the signal has not yet reached saturation, this should be determined experimentally by testing multiple cargo doses and incubation times. Saturation occurs when additional uptake of particles contributes to a weaker increase in fluorescence signal, so will be affected by instrument sensitivity, resolution and dynamic range [154]. Saturation will be achieved more rapidly if the ratio of cargo particles to microglia is high, and cargo particles are small [155]. Other factors that affect the phagocytic capability of the in vitro microglia cell model should also be considered and controlled, such as the inclusion or exclusion of serum from the media [153], and the duration of in vitro culture prior to assaying [156]. Appropriate negative controls should be used to determine the amount of specific phagocytosis measured versus background signal. Pre-incubating the cells with actin cytoskeleton inhibitor cytochalasin D and maintaining it in the media should inhibit phagocytosis by approximately 90% [138].

The accuracy and reproducibility of the data can be improved by sampling more cell events or capturing more image fields. However, if live cells are measured then increasing the sampling could lead to unacceptably long delay in data capture. Therefore, sampling depth needs to be balanced by speed. Instruments with higher levels of automation can improve processing speed and additionally have the benefit of reducing operator bias. Image-based methods with the ability to resolve intracellular structures (recommend 40X magnification or higher) allow for more phagocytosis parameters to be measured, which can improve detection accuracy.

Finally, consideration should be given to the choice of fluorescent labels in the assay. To allow intracellular events to be detected, the microglial cells are usually stained with a fluorescent chemical or lectin dye to highlight the whole cell body, or else they are fluorescently labelled with an antibody for a microglia-specific marker after phagocytosis has occurred. For high-content imaging, microglia staining is particularly important to ensure that cells are ‘segmented’ accurately in the automated image analysis pipeline. Furthermore, cargo can be labelled prior to phagocytosis with a permanently fluorescent dye (e.g., Alexa Fluor-488) or a pH-sensitive dye (e.g., pHrodo iFL Red). pH-sensitive dyes are weakly fluorescent at neutral pH and increase their fluorescence with reduced pH, such as occurs with phagosome acidification [157]. This can improve discrimination of phagocytosed cargo from external membrane-bound cargo, however the user needs to be aware that defective phagosome acidification can affect the signal of a pH-sensitive dye, in addition to the mechanics of phagocytic uptake. This phenomenon can be exploited by co-labelling cargo with pHrodo Red and Alexa Fluor488 in order to assay phagosomal pH [158]. Permanently fluorescent dyes are not altered by endosome acidification, therefore distinguishing between external bound particles and phagocytosed particles is more challenging. This can be aided by the addition of trypan blue to quench extracellular fluorescence [159].

To conclude, phagocytosis assays can provide important insights into the effect of chemical or genetic manipulations upon microglia function. Phagocytic cargo should be chosen carefully with the intended biological question in mind, and ideally several types should be tested. Flow cytometry and imaging readouts are commonly used, and robust data can be obtained from relatively inexpensive equipment if the assay is carefully optimised. However, high-content imaging is advantageous when the perturbation of phagocytosis is expected to be subtle, because it more accurately quantifies the amount of phagocytosed material per cell.

## 7. Chemotaxis and Random Mobility

### 7.1. Overview of Chemotaxis and Random Mobility

Microglia are highly dynamic cells that are constantly in motion. In their resting state, healthy microglia extend and retract their processes constantly to survey their local neuronal network, whilst largely retaining their cell body in the same position and maintaining distance from other microglia [4]. This “surveillance behaviour” is a form of undirected motility and governed by mechanisms independent from chemotaxis [160]. Chemotaxis is a specific form of directed motility whereby cells move towards or extend their processes towards the emitter of a secreted chemoattractant molecule, up a concentration gradient. The term is often used to more broadly encompass all forms of directional migration [161], and will be used interchangeably with directional migration here. Brain lesions caused by trauma, infection, or neurodegenerative disease result in the localised production of chemoattractants CCL21 and CX3CL1 by neurons and astrocytes, and MCP-1/CCL2 and SDF-1α/CXCL12 by activated microglia adjacent to the lesion [162]. Microglia in the vicinity of the damaged region initially just extend their processes towards these signals, and over a period of many hours some microglia cell bodies migrate to the injury site, with the aim of containing and clearing the damage [4,163]. Damaged cells also release purines such as ATP, ADP and UDP, which strongly promote microglia motility but cannot be strictly considered as chemoattractants because their effect is not directional, they only increase the speed of movement (in any direction), which is known as chemokinesis [164].

### 7.2. Functional Assays for Chemotaxis

Methods of assaying macrophage chemotaxis in vitro have been reviewed extensively elsewhere, and would apply to microglia [161,165]. Here, we will focus on the chemotaxis assay most commonly used in microglia research: transwell assays. Other methods such as the Dunn chemotaxis assay [166] and microfluidics [167] have been used on microglia and are also worth considering. These are excellent methods for assessing chemotaxis and distinguishing chemotaxis from other types of cell motility, but have the limitation of being very low-throughput, only allowing one or two conditions to be tested [161,165]. True microfluidic set-ups are also custom-made for each experiment, and require access to 3D-printing equipment [165].

#### Transwell Assays

Transwell assays are adapted from the original Boyden chamber, which is a well containing chemoattractant solution, with cells placed in an insert with a porous membrane base that is in contact with the chemoattractant [165]. Due to the small size of the membrane pores, a steep chemoattractant gradient is maintained for hours, but the microglia are able to migrate through the pores and onto the underside of the insert. In a traditional fixed endpoint transwell assay (Figure 3—Transwell assay workflow), the cells are allowed to migrate for a set period of time and then fixed and stained with dyes for easy visualization, e.g., Hoechst [123] or crystal violet [166]. It is possible to monitor kinetics in specially modified 96-well plate transwell set-ups, such as the IncuCyte chemotaxis assay and the xCELLigence impendence assay [165,168].

In microglia literature the transwell assay is by far the most popular chemotaxis method. Mouse microglia migration towards ATP [166], ADP [166], CCL2 [169], CCL19 [140], CCL21 [140], Tau [170], and α-Synuclein [171] has been assayed by this method, amongst numerous other examples. Human microglia including iPSC models have also been tested, with ADP [123,138], C5a [138], and Aβ1-42 [172] as ‘chemoattractants’. The advantages of transwells are that they are the most scalable method currently available and can be performed in 96-well plates. Transwell assays are easy to use, simple to analyse, and require minimal optimization. However, the fixed endpoint assays do not provide kinetic information and have very limited throughput. Microglia are lifted from their culture plate immediately prior to the assay, which may alter their activation state and behaviour. Furthermore, it is not easy to reliably distinguish chemokinesis (increased non-directional motility) from chemotaxis with the transwell method. Researchers have tried to diagnose chemokinetic versus chemotactic behaviour via a ‘checkerboard analysis’, but this method is considered to be extremely flawed by some (as discussed in [161]). Therefore, to ascertain whether a molecule-of-interest is a true chemoattractant, it would be advisable to validate with a Dunn chemotaxis assay or under-agarose assay or microfluidic set-up.

### 7.3. Functional Assays for Random Motility

The undirected or ‘random’ motility of microglia is important for their homeostatic surveillance of the brain environment. Random motility employs similar downstream cytoskeletal effectors to chemotaxis, however the upstream signalling appears to be distinct, being P2Y12-independent [160]. Therefore, they are not a substitute for chemotaxis assays, but can provide complementary information about the effect of an intervention on microglia motility. Here, we consider two methods to study non-directional motility: direct imaging, and scratch assays/exclusion stamps.

#### 7.3.1. Direct Imaging

Direct time-lapse imaging of cultures can be used to track the movements of individual cells and cell processes moving randomly in culture. Microglia are visualised with a fluorescence reporter or live-cell stain. Monocultures can be monitored, however microglia motility is blunted in the absence of neurons [173], so the method is more effective in vivo or with more complex in vitro models, e.g., co-cultures of microglia with neurons. Analysis of baseline process motility can be performed on binarised images with custom MATLAB scripts, essentially examining all process additions and process retractions of a microglia between pairs of consecutive time points [160,174]. More simple tracking of cell soma movements can be performed manually with ImageJ [173]. Alternatively, some image analysis software such as Imaris are capable of automated cell soma tracking and process detection, and has been used for in vivo and in vitro microglia [175,176]. The advantages of this technique is that it truly measures microglia surveillance behaviour with little disturbance, and the output can be rich in detail with many single-cell parameters measured. The limitations are that it requires automated time-lapse microscopy, and is data-intensive and time-consuming to analyse, even with good cell-tracking software.

#### 7.3.2. Scratch Assays/Exclusion Stamps

Wound-healing scratch assays involve a fine linear scratch being drawn through a dense monolayer of cells in culture, and live imaging is used to monitor the width of the scratch as cells migrate into it, or the number of individual invading cells. Scratch assays may damage cells, but no chemotactic gradient is sustained in the well, therefore closure of the wound is governed by random motility [177]. Scratching of cultures is usually done with a plastic pipette tip and a steady hand, however more reproducible scratches in 96-well format can be achieved with an IncuCyte WoundMaker tool [176]. Exclusion stamps (e.g., iBidi, Oris) are a variation of the scratch assay method, where a silicon stamp is inserted into a well prior to cell plating, and removed after cell attachment has occurred, leaving a space for the cells to randomly migrate into [178]. The advantages of scratch and exclusion-stamp assays are that they do not require specialist equipment or software, can be adapted for high-throughput screening, and are an effective model of undirected migration in microglia monocultures, which are likely to be fairly static at equilibrium [177]. The main limitations are that a larger number of cells are needed relative to other methods, at a much higher density than is physiological, and several days of imaging are required [177].

In conclusion, microglia motility can be chemotactic–directed towards a stimulus–or random/undirected. Easy medium-throughput assays are available for measuring chemotaxis and random motility–transwells for the former, and scratch assays for the latter–but these have limitations that should be weighed up. A more detailed study of motility should include multiple complementary assays, and consideration given to the biological relevance of any stimuli. Kinetic information is particularly important for motility studies, therefore it is advantageous to use a good time-lapse imaging microscope with environmental control.

## 8. Microglial Immunometabolism

### 8.1. Overview of Microglial Immunometabolism

Microglia are highly plastic cells, displaying rapid physiological changes in response to a wide variety of external stimuli. Functionally, these changes include gross morphological changes, release of inflammatory and anti-inflammatory molecules, phagocytosis, endocytosis, chemotaxis and migration to sites of injury. Executing these processes requires energy from metabolism [179,180,181]. The dependence of immune functions on cellular metabolic pathways can be described using the term immunometabolism. Alongside changes in metabolism, activation of microglia can result in changes to mitochondrial structure, with LPS treatment shown to result in mitochondrial fragmentation, indicative of reduced OXPHOS [182,183,184].

Microglia are capable of generating energy via both glycolytic and oxidative metabolism [185,186]. These cells are able to perform rapid ‘immuno-metabolic switching’ from oxidative metabolism to glycolysis in order to meet a sudden increase in energy demand [179,181,187]. Oxidative phosphorylation (OXPHOS) involves shuttling of the pyruvate generated from glucose to mitochondria, where it is converted into acetyl coenzyme A, and enters the tricarboxylic acid cycle. Glycolysis begins with the uptake of glucose into cells via various glucose transporters [54,188,189]. This glucose is then enzymatically converted to pyruvate, then lactate, in order to generate adenosine triphosphate (ATP). Glycolysis occurs considerably faster than OXPHOS, but is much less efficient: generating 20-fold less ATP. For this reason, glycolysis is often used by cells in response to a high, transient energy demand [179,181].

For microglia, increased demand often occurs following recognition of an external stimuli, such as DAMPs, PAMPs, or NAMPs [180]. ATP is required for reorganization of the actin cytoskeleton: critical for cell morphological changes, phagocytosis and cell migration [190]. Furthermore, other by-products of metabolism such as ROS, fatty acids, amino acids and nucleotides are required for phagocytosis and the production of cytokines [191].

Sustained glycolysis results in increased mitochondrial membrane potential (ΔΨm), and likewise increased reliance on OXPHOS results in a decrease in ΔΨm [192]. In macrophages, this has been suggested to occur because when cells are relying on glycolysis to generate ATP, protons generated in the electron transport chain are no longer being used by mitochondrial ATP synthase during OXPHOS [192,193]. While ΔΨm is for the most part highly regulated by cells, defects in this parameter can have detrimental consequences regarding immunometabolism and cell health [194].

### 8.2. Functional Assays for Immunometabolism

A wide variety of assays exist that can be used to examine mitochondrial function in vitro, which have been reviewed extensively elsewhere [195,196,197]. The most common methods will be discussed below.

The Agilent Seahorse XF analyser acts as an incredibly powerful tool for examining mitochondrial function [195,196]. This machine works by using solid-state probes at the bottom of a custom 96-well plate to measure both levels of O2 and pH within cell media in real time. From these measurements, the Seahorse can calculate both oxygen consumption rate (OCR) and extracellular acidification rate (ECAR) within a culture. Kits are available which allow the study of specific subsets of cellular metabolism, e.g., the Mito Stress Test measures a variety of parameters related to OXPHOS, and the Glycolytic Stress Test, which does the same for glycolysis. These kits work by providing a variety of respiratory modulators, which are added to cells sequentially by the machine whilst OCR and ECAR measurements are taking place. When considering both equipment and reagent costs, Seahorse assays can be expensive to run. However, data obtained is robust and multiparametric, and the analysis software is easy to use. Furthermore, assays can be easily personalised to reflect specific research questions. Practical considerations when performing Seahorse assays include data normalization to accurate cell counts: small differences in cell number can cause significant differences in readings. In addition, care must be taken to ensure all media and modulators are at neutral pH prior to the assay, as the assay is sensitive to small changes in pH.

An additional method to examine mitochondrial function in vitro is to assay ΔΨm [197]. Several fluorescent probes exist for this purpose, with the most commonly used being tetramethylrhodamine methyl ester (TMRM) [198]. Most ΔΨm probes are substrates for multidrug resistance transporters on the mitochondrial membrane, and their level of accumulation in mitochondria is directly proportional to ΔΨm. While a marked decrease in ΔΨm can indicate poor cell health, it can also indicate reduced OXPHOS [192], which should be assessed with an additional assay, e.g., Seahorse.

Immunometabolism can be indirectly assayed by visualizing mitochondrial network structure, using fluorescent staining (e.g., Mitotracker), immunocytochemistry or electron microscopy [197]. Within cells, the mitochondrial network constantly undergoes fusion and fission events, reflecting changes in their function and status [199]. For this reason, identification of mitochondrial morphology followed by characterization of the size and shape of these organelles can provide significant information regarding their bioenergetics.

Immunometabolism can be more comprehensively assessed with metabolomics. Metabolomics characterises and quantifies the metabolome: the total set of metabolites, substrates, intermediates and products of cellular metabolism. This process requires the use of mass-spectrometry in conjunction with liquid or gas chromatography and/or nuclear-magnetic resonance. While this technique is expensive and requires specialist equipment and expertise, it generates a large quantity of data about the current state of metabolic processes within a culture. However, it should be noted that metabolism is a highly dynamic process, and metabolomics only provides a snapshot at one point in time [195].

The importance of immunometabolic switching with regard to microglia function highlights the value of studying this process when investigating microglial phenotypes in vitro. Using more than one method to assess mitochondrial function in vitro is advised for more robust conclusions.

## 9. Conclusions

It is clear that microglia exhibit numerous important and complex functions within the brain. Fortunately, a large number of well-developed methods exist by which to measure each of these functions in vitro. When choosing which method to use for assessing each function, it is important to take into consideration the research aims.

Upon selecting stimuli or cargo, the most relevant choice should be used. In addition, future studies should aim to move the field towards more physiologically relevant stimuli when studying microglia activation. Broadly, accuracy and reproducibility of the data can be improved by having sufficient technical replicates, and sampling more cell events for flow cytometry, or capturing more image fields for microscopy readouts. Moreover, performing multiple assays for one or more related functions can provide a more nuanced picture of the effect of an experimental perturbation.

We hope that this review will provide microglia researchers with all the information they require to perform a comprehensive evaluation of this highly important cell type.

## Figures and Tables

**Figure 1 cells-11-03414-f001:**
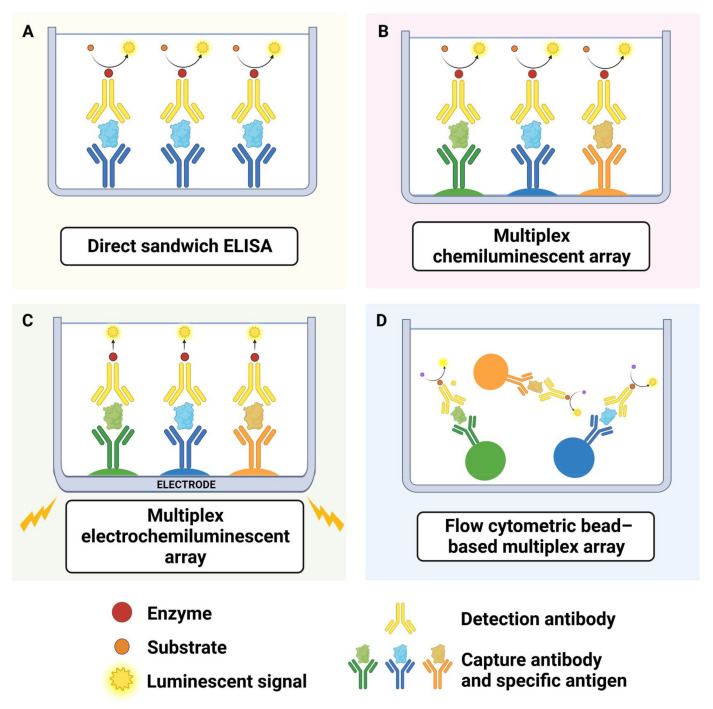
Overview of arrays for quantifying cytokine release. (**A**) ELISA: a 96 well plate is coated with a capture antibody specific to an antigen within the protein of interest, e.g., one specific cytokine/chemokine. Next, staining with another antibody that binds to the protein of interest and that is conjugated to an enzyme (e.g., streptavidin/biotin) allows detection/quantification of protein (Kohl and Ascoli, 2017). (**B**) Chemiluminescence arrays: multiple specific biotin-conjugated detection antibodies are added to each well, thus allowing quantification of multiple cytokines following the addition of horseradish peroxidase-conjugated streptavidin and measurement of luminescence. (**C**) Electrochemiluminescence arrays: each detection antibody, instead of being conjugated to biotin, is conjugated to a proprietary tag that is excited with emission beams in the electric field. (**D**) Bead based multiplex arrays: use proprietary bead sets that can be distinguished from each other via flow cytometry (due to varying size/fluorescence of bead types). Each type of bead comes conjugated to an antibody specific to one of the proteins of interest. Next, either streptavidin or fluorescence labelled antibodies are added that bind specifically to each cytokine-antibody complex on the bead sets. Using a flow cytometer, up to 25 cytokines in the same sample can be measured with commercial kits; or up to 100 with custom-conjugation [84].

**Figure 2 cells-11-03414-f002:**
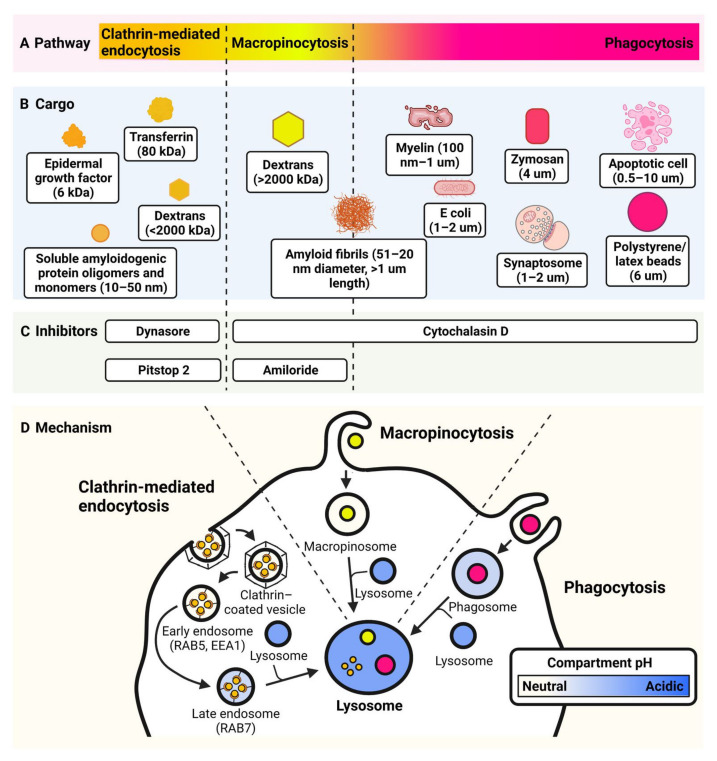
Schematic of methods for assaying endocytosis and phagocytosis. (**A**,**B**) Pathways and cargoes of endocytosis. The size of particle determines the pathway utilised, indicated by colour on the schematic. Clathrin-mediated endocytosis (<120 nm) is shown in orange. Macropinocytosis (>200 nm) is shown in yellow. Phagocytosis (>500 nm) is shown in pink. (**C**) Inhibitors used in endocytic assays. (**D**) Mechanisms of endocytic pathways. Blue indicates relative acidity of compartments. The lysosome is pH 4.5–5.0. Endosomal protein biomarkers are also indicated.

**Figure 3 cells-11-03414-f003:**
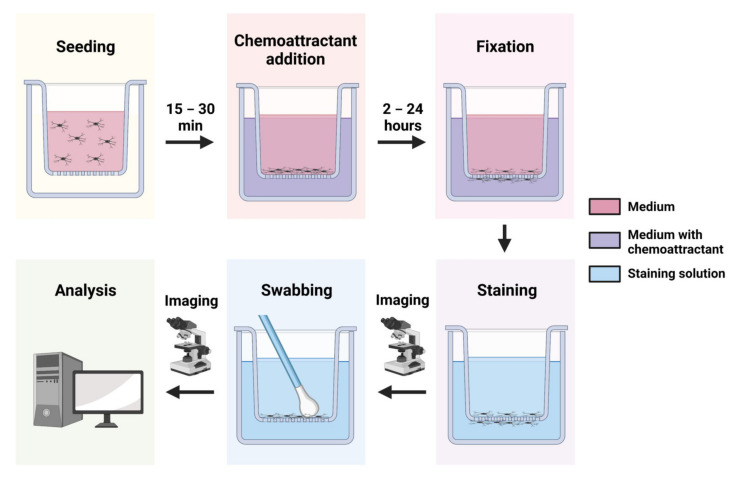
Transwell assay workflow. Microglia are seeded into transwells and can be left to adhere to the membrane for up to 30 min. A defined chemoattractant is then added to the bottom chamber and microglia are allowed to migrate through the membrane for a defined timespan. Cells are fixed and stained. Light microscopy at a low magnification is used to image the inserts allowing visualization of all cells either side of the membrane. The inserts are then swabbed on the top side with a cotton wool bud to remove the unmigrated cells, and imaged again to visualise migrated cells only. Image analysis software is used to count cells, and the result is expressed as a ratio or percentage of migrated cells (after swabbing) relative to the total cell population (before swabbing).

**Table 1 cells-11-03414-t001:** Common microglia identity markers. TMEM119—Transmembrane protein 119; IBA1—Ionized calcium binding adaptor molecule 1; P2RY12—P2Y Purinergic receptor 12; CD11B—Cluster of differentiation 11B; CD45—Cluster of differentiation 45; CD68—Cluster of differentiation 68; TREM2—Triggering receptor expressed on myeloid cells 2; PU.1—Transcription factor PU.1; C1QA—Complement Component 1 subunit Q A chain; GPR34—G protein coupled receptor 34; MERTK—MER proto oncogene tyrosine kinase; CX3CR1—C-X3-C chemokine receptor 1; SALL1—Sal like protein 1; GLUT5—Glucose transporter 5; GAS6—Growth arrest specific 6; CD14—Cluster of differentiation 14.

Marker	Function	Expression in CNS Cell Types and Microglial Models	Expression after Proinflammatory Stimulation	References
TMEM119	Currently uncertain	Microglia. Highly conserved. Low mRNA in some iPS models.	Downregulated	[26,30,39]
IBA1	Binds calcium and actin supporting the cytoskeleton and membrane ruffling	Microglia.	Upregulated	[40]
P2RY12	Detects nucleotides such a ATP following injury	Microglia and oligodendrocyte precursor cells. Low mRNA in some iPS models.	Downregulated	[26,38,41]
CD11B	An integrin for complement receptor 3 (CR3), used for phagocytosis of complement-coated cargo	Microglia. Some have shown to be absent on the HMC3 cell line.	Upregulated	[34,42,43]
CD45	Receptor protein tyrosine phosphate involved with T cell signalling and proliferation	Microglia.	Upregulated	[34,35]
CD68	Lysosomal marker	Microglia. Some have shown to be absent on the HMC3 cell line.	Unchanged	[26,42,44]
TREM2	Key in development and maintenance of the brain with a role in synaptic pruning and immune response	Microglia.	Downregulated	[34,39,45,46]
PU.1	Transcription factor determining myeloid lineage	Microglia.	Unchanged	[47]
C1QA	Opsonin that triggers the classical complement cascade	Microglia.	Upregulated	[26,48]
MERTK	Transduces signals, involved in cell survival, migration and phagocytosis	Microglia, astrocytes, rod photoreceptor cells and oligodendrocyte precursor cell.	Downregulated	[39,49,50,51]
CX3CR1	Involved in the immune response, inflammation, cell adhesion, chemotaxis and migration	Microglia.	Upregulated	[26,52]
SALL1	Transcriptional master regulator of microglia identity and non-inflammatory functions	Microglia, oligodendrocytes, and astrocytes.	Downregulated	[27,39,53]
GLUT5	Fructose transporter	Microglia and some neuronal populations.	Upregulated	[54]
GAS6	Simulates cell proliferation and shown to blunt inflammatory response of LPS	Microglia, fibroblasts and astrocytes.	No information available	[55]
CD14	Co receptor for TLR4 and TLR7/9	Microglia.	Upregulated	[26,56]

**Table 2 cells-11-03414-t002:** Summary of the different techniques used to quantify cytokine release from microglia, as well as pros and cons of each technique and examples of products. ELISA = enzyme-linked immunosorbent assay.

Type of Assay	Max Cytokines/Assay	Pros	Cons	Examples
Double antibody sandwich ELISA	1	Most widely used & best validated, highly quantitative, reproducible	Measures only a single protein/sample, dynamic range narrow in relation to other cytokine assays	R&D Systems^®^ ELISA Kits, Abcam ELISA Kits, Invitrogen ELISA kits
Multiplex chemiluminescent arrays	9	Efficiency (time & cost), easier to do multiple timepoints, higher dynamic range than traditional ELISA, less sample needed, allows evaluation of one inflammatory molecule in the context of multiple others	May require specialist paid software to analyse results (e.g., Quansys Q-plex™ requires Q-view software)	Quansys Q-Plex™ (Oxford Biosystems), Luminex^®^ Multiplex Assays (Thermofisher), Multiplex Immunoassays (Bio-Rad)
Multiplex electrochemiluminescence arrays	10	Efficiency (time & cost), lack of enzymatic or fluorescent detection system avoids time-dependent signal decay, less sample needed, allows evaluation of one inflammatory molecule in the context of multiple others	Expensive, requires specialist equipment	Meso Scale Discovery (MSD)
Flow cytometric bead-based multiplex arrays	100	Efficiency (time & cost), compatible with standard flow cytometers, allows evaluation of one inflammatory molecule in the context of multiple others, free analysis software, cheapest method per sample/cytokine, easier to do multiple timepoints, less sample needed	Less tested compared to traditional sandwich ELISA	BD Cytometric Bead Array (BD Biosciences), LEGENDplex™ (BioLegend)

**Table 3 cells-11-03414-t003:** Comparing types of instrument used to measure phagocytosis. +: low; ++: medium, +++: high; ID: Instrument-dependent; N/A: not applicable; Y: yes.

	Spectro-Photometer	Fluorescence Microscope (No Automation)	Time-Lapse Fluorescence Microscope	High-Content Imaging System	Flow Cytometer	Imaging Flow Cytometer
Sensitivity	+	ID	++	+++	+++	+++
Magnification	N/A	ID	++	+++	N/A	++
Multiple data parameters	+	ID	++	+++	+	++
Unbiased data collection	+++	+	++	+++	+++	+++
Unbiased data analysis	+++	+	++	+++	++	++
Speed	+++	+	++	+++	+	+
Cell viability	++	++	+++	+++	+	+
Real-time kinetics	Y	N/A	Y	Y	N/A	N/A
Single-cell analysis	N/A	Y	ID	Y	Y	Y
Example instrument	SpectraMax (Molecular Devices)	LSM700 (Zeiss)	Incucyte ZOOM (Sartorius)	Opera Phenix (Perkin Elmer)	FACS Calibur (Becton Dickinson)	Amnis ImageStreamX Mk II (Luminex)
Assay references	[151]	[152]	[153]	[138,144]	[133,145]	[123,137]

## Data Availability

Information on the data underpinning the results presented here, including how to access them, can be found in the Cardiff University data catalogue at http://doi.org/10.17035/d.2022.0228053760.

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
