# Peer review of "Assaying Microglia Functions In Vitro"

_cells, 2022, doi:10.3390/cells11213414_

Round 1

Reviewer 1 Report

The review by Maguire et al. provides an excellent overview of functional assays for cultured microglia. The authors discuss assays to assess inflammatory responses to distinct stimuli, migration, endocytosis/phagocytosis, motility, and immunometabolism. I have only minor suggestions that may be incorporated into a revised version of the manuscript:

- In the introduction, the authors should briefly mention differences between in vivo and cultured microglia with respect to their transcriptome, morphology, and maybe functional differences, e.g., response to inflammatory stimuli.

- The endocytosis section includes aspects of phagocytosis, e.g. with regards to Abeta. Either the authors merge endocytosis and phagocytosis, or should clearly keep those sections separate.

-

Author Response

All authors would like to thank you for your constructive feedback to improve our manuscript. We have now updated the manuscript to reflect your comments and relevant altered sections taken from the manuscript are shown below highlighted in yellow.

We appreciate your comment regarding the discussion of Aβ uptake in the endocytosis section and have updated the section to reflect this (line 323, & 333-334 – rearrangement of text). However, the authors feel it is important that the discussion regarding Aβ is kept in the endocytosis section given that a large proportion of Aβ uptake does not occur through phagocytosis but instead by other endocytic mechanisms. We feel this is important to highlight in the field to stop confusion and help others select the correct assay depending on the form of Aβ being assayed.

With regards to your second comment, suggesting a discussion of the differences between in vivo and cultured microglia, we have edited the text to include more information. (lines 31-33): Microglia, which make up ~0.5-16.6% of the total cell population in the human brain [1], display a great deal of heterogeneity in vivo regarding age, sex, and location within the CNS. This includes variation in cell density, morphology, and function [1-3]. With respect to differences in morphology however, we believe that this is already covered in the morphology section where we state (lines 153-154): ‘Furthermore, it should be noted that isolated microglia never achieve the morphological complexity of microglia observed in situ’.

In addition to your comments, following submission we noticed that in Figure 2 we missed out labels highlighting parts A, B, C, and D, and have edited the figure to correct this.

Again, we would like to thank you for your valuable contributions to our manuscript. We hope our amended manuscript will now be accepted for publication.

Reviewer 2 Report

The paper is well written and contain relevant information to people working on the field, specially on cell culture.  I have some suggestions to increase the quality of the paper.

General suggestion: the authors should standardise the language. Currently, the paper presents a mix of American and British English.

1-Introduction

- line 31-32. The authors should expand the information about microglial distribution in the brain. There is a solid literature about the heterogeneity of microglial density according to brain area, age and sex, for example.

2-Markers for microglia

- line 66. The authors suggest that TMEM119 is the most used microglial marker as they say right in the next sentence “a second widely used marker is P2YR12…”. I suggest they discuss a bit more IBA1, which is arguably the most used microglial marker specially in the healthy brain. Also, it is highly recommended for quantifying morphological remodelling as it shows even fine ramification. So given the fact that IBA1 is unique to microglia in intact brain and represent one of the most complete view of microglial morphology, the authors should spend a bit more lines about it.

3- Morphology

- the authors could briefly discuss the wide range of microglial morphotypes related to age, physiological and pathological contexts.

Author Response

All authors would like to thank you for your constructive feedback to improve our manuscript. We have now updated the manuscript to reflect your comments and relevant altered sections taken from the manuscript are shown below highlighted in yellow.

As per your comment, we have updated the language to be consistent British English throughout. In addition, we have added more information about microglial distribution in the brain (lines 52-56) In brief, in vitro microglia appear more “activated”, secreting a greater number of cytokines when compared with their in vivo counterparts. Furthermore, the transcriptomic signature of in vitro microglia differs significantly from in vivo models. These changes likely arise following a lack of signalling between microglia and other CNS cells, which helps microglia retain a more “homeostatic” phenotype [22,23].

Also, further information about the microglial marker IBA-1 (lines 72-80) has been added. Additional popular microglia markers include IBA-1 and P2RY12. Both of these markers are expressed by microglia and macrophages, with expression of P2RY12 on macrophages shown to be much lower than microglia expression [29]. Indicating that neither of these markers can be used to distinguish between microglia and macrophages [30]. Regardless of this IBA-1 is a useful marker for the visualisation of microglia morphology, as staining extends through the slender protrusions of ramified microglia [31]. Furthermore, IBA-1 expression has been shown to be upregulated during phagocytosis and migration processes due to its interaction with actin which has led it to be considered a good marker for the initial stages of inflammatory microglia activation [32]. We also edited Table 1 to give IBA-1 a more prominent position, reflecting the information in the main body text.

We have also included additional information regarding the wide range of microglial morphologies (lines 120-122). Microglia morphology can be seen to change drastically during both physiological and pathological ageing, as well as in response to a variety of different neurological conditions [58,59].

In addition to your comments, following submission we noticed that in Figure 2 we missed out labels highlighting parts A, B, C, and D, and have edited the figure to correct this.

Again, we would like to thank you for your valuable contributions to our manuscript. We hope our amended manuscript will now be accepted for publication.